# Placebo and Nocebo Effects on Sports and Exercise Performance: A Systematic Literature Review Update

**DOI:** 10.3390/nu16131975

**Published:** 2024-06-21

**Authors:** Bhavya Chhabra, Attila Szabo

**Affiliations:** 1Institute of Health Promotion and Sport Sciences, Faculty of Psychology and Education, ELTE Eötvös Loránd University, 1075 Budapest, Hungary; bhavyac@student.elte.hu; 2Doctoral School of Education, Faculty of Psychology and Education, ELTE Eötvös Loránd University, 1075 Budapest, Hungary; 3Faculty of Sport and Health Sciences, Széchenyi István University, 9025 Győr, Hungary

**Keywords:** ergogenic aids, nutritional, mechanical, placebo, nocebo, performance

## Abstract

Sports performance could be affected by placebo and nocebo effects. The last literature review on placebo and nocebo effects on sports and exercise performance was published in 2019. In the past five years, several new studies have been published. This review aimed to update the previous synthesis and evaluate the results of new studies focusing on placebo or nocebo interventions in sports and exercise by determining the form and magnitude of their effect. Hence, we searched for empirical studies published from 2019 until the end of May 2024 indexed in PubMed, Medline, Web of Science, EBSCO, and Google Scholar databases. The search yielded 20 eligible studies with control or baseline-control conditions, focusing on nutritional, mechanical, and other mixed ergogenic aids. They yielded small to large placebo effects (Cohen’s *d*) for nutritional (*d* = 0.86), mechanical (*d* = 0.38), cream and gel (*d* = 0.05), and open-label placebo (*d* = 0.16) interventions. The pooled effect size for placebo effects was moderate to large (*d* = 0.67), larger than in the earlier review, suggesting that placebo effects can improve motor performance even more than previously reported. However, based on five measures from three studies, the nocebo effects were almost twice as large (*d* = 1.20). Accordingly, the current findings support and expand the last review in the field by yielding additional support for placebo and nocebo effects in sports and exercise.

## 1. Introduction

A placebo is an inactive agent, event, or idea not expected to have systemic effects. Its effects surface from a person’s expectations or conditioned earlier experiences [1]. The word placebo comes from Latin. The Merriam-Webster dictionary defines it as “*I shall please*”, while its opposite, nocebo, is interpreted as ‘*I will be harmful*’. While learning is instrumental in placebo and nocebo effects, the strength of belief, determined by personal, situational, and certainty factors, determines their impact on the outcome [1].

Placebo and nocebo effects mirror the remarkable power of the human mind to shape future outcomes [1]. Hill [2] elaborates on how one can perform and engineer one’s own ‘miracles’ through transforming thoughts and beliefs. In this regard, the placebo effect can enhance performance by creating positive expectations, while, on the other hand, the nocebo effect would hinder one’s performance through negative expectations [1,3,4,5]. These effects have been studied in medicine for over five decades for their notable neurobiological and psychological effects [1,4,6,7]. Similar to Szabo’s [1] model, Clark [8] purports that the placebo effects characterize a ‘favorable outcome’ resulting typically from the ‘belief’. This belief activates neurobiological mechanisms in the brain, leading to the expected outcome [6,9]. The expectancy later, shaped by classical conditioning, is the route of the placebo effect, and vice-versa, the negative expectancy protocol results in the nocebo effect [10]. These effects have recently received increased attention in sports and exercise psychology due to their strong association with motor performance [5,11]. Based on this connection, some authors [1] consider placebo effects as a form of covert doping. Therefore, understanding and estimating the potential strength of this undetectable and often deception-laden mental manipulation is essential concerning athletic performance.

### 1.1. Placebo and Nocebo Effects in Sports

In sports science, understanding the placebo effect and recognizing its underlying mechanisms has significant implications for research and practice. First, it influences how the experiments are structured to measure the magnitude of the placebo effect linked with treatments [1,5]. In this regard, it is essential to consider its ergogenic effects within the psychological or social context. Second, it influences the independent evaluation of both placebo types and their effectiveness. Third, it expands our understanding of the placebo effect’s role in sports [5]. Studying the effects of nocebo is essential for determining what should be avoided in sports and exercise to prevent performance deterioration [5].

Empirical studies on placebo ergogenic aids, such as anabolic steroids, caffeine, cold water immersion, various gels, ischemic preconditioning, amino acids, mints, pills, verbal suggestion, transcutaneous nerve stimulation [TENS], and the like, have started in the new millennium. So, this is a relatively young but fast-growing field. For example, Hurst et al. [11] found 31 articles in 19 years between 2000 and 2018 (plus one published before the millennium), yielding an average of 1.6 articles per year. In the current review, we located 20 papers (see below), representing a two-and-a-half times (4.0) increase. However, reviews only include *intervention studies* that are difficult to conduct because they most often involve deception. In terms of general interest in the topic, based on the ‘*placebo or nocebo*’ and ‘*sport or exercise*’ word pairings in the titles of the articles in PubMed, another picture emerges, showing that publications peak in 2020–2021 with a decline in 2022–2023 (Figure 1). We hope our review will generate revived interest in this exciting but controversial topic.

Research has shown considerable improvement in the sporting performance of athletes [11]. For example, Duncan [12] found that participants exhibited a 23.8% enhancement in weight lifted during a leg extension task when they received a caffeine placebo. Likewise, Kalasountas et al. [13] reported an 11% rise in weightlifting after participants consumed a placebo believed to contain amino acids. Beedie et al. [14] found that negative information associated with a fictive supplement decreased running speed. Such adverse effects may be the result of disinterest or anxiety [10].

### 1.2. Scope of Review

This review aims to expand the review conducted by Hurst et al. [11] on placebo and nocebo effects in sports and exercise. Thus, the objective is to offer an updated synthesis of the newly emerging work, covering the period from 2019 to May 2024.

## 2. Materials and Methods

This review was pre-registered on the Open Science Foundation (OSF) Registries (https://doi.org/10.17605/OSF.IO/A3N6D). The work follows the Preferred Reporting Item for Systematic Review and Meta-Analysis (PRISMA) guidelines [15,16].

### 2.1. Type of Studies

Intervention studies published between 1 January 2019 and 31 May 2024 were included in this review if they examined placebo and/or nocebo effects on various motor performances and met the inclusion criteria (Table 1). The literature search targeted the English-language empirical studies in peer-reviewed journals and those that included both within-participant and between-participant research designs and contained control or baseline-control conditions. Briefly, English peer-reviewed papers using a placebo or nocebo intervention that could be replicated and had a control measure to which a placebo or nocebo effect could be contrasted were deemed eligible for inclusion.

### 2.2. Type of Participants

Participants in any form of sport or exercise were considered. Studies examining recreational exercisers, athletes, or healthy individuals were considered for inclusion. We also included studies with healthy paralympic individuals and people not diagnosed with an organic or mental dysfunction, such as children affected by obesity.

### 2.3. Type of Interventions

We considered all studies that could establish differences (in comparison to a control group) or changes (in comparison to a baseline) attributable to their intervention. Further, we only included interventions that could not have an isolated or independent effect on the target measures. The placebo or nocebo could have nutritional, mechanical, or other characteristics, including verbal manipulation.

### 2.4. Outcome Measures

Studies that lacked at least one objective motor skill-related performance measure, for instance, speed, time to completion, or power output, were excluded. Furthermore, we excluded studies that reported subjective measures such as perceived exertion, fatigue, or similar measures without additional objective motor performance measures.

### 2.5. Interventions

We excluded studies in which the intervention effects could also be attributed to something other than placebo or nocebo effects (e.g., before conducting the review, we decided that we would not include works in which participants could be aware of receiving either a placebo or nocebo unless the research uses an open-label placebo).

### 2.6. Search Strategy

Study selection occurred in two stages: (a) reviewing titles and abstracts and (b) examining full texts. We searched the PubMed, MEDLINE (through the PubMed interface), Web of Science, and EBSCO databases. Further, we also searched the Google Scholar database, which, six years ago already, located 95% of citations in 252 subjects [17]. Furthermore, if a study was eligible based on its abstract, we read its full text and consulted its references to locate further eligible studies. We used clusters of terms summarized in Table 2 during the literature search with two Boolean operators (‘OR’ for widening the search and ‘AND’ for narrowing the search). However, we performed an ‘advanced search’ using the combination of words, forming the left and right columns of Table 2 in Google Scholar (Table 3).

There was no age, geographical, or cultural constraint during the search. Two researchers searched the databases, and upon completion, they compared and discussed the obtained records. Subsequently, they decided whether to read the full text of the paper or not. After reading the complete text, the authors assessed the matched articles with the inclusion criteria.

### 2.7. Data Collection and Analysis

All eligible studies were extracted and summarized in Table 4, which includes details of the studies, including author names, the year of publication, study design, participants, dependent variables, type of placebo or nocebo treatment, the information provided to participants, and significant study findings. In Table 4, in the last column, the studies’ key findings are presented briefly; when necessary, we elaborate on results in the text based on categorizations and subcategorizations.

### 2.8. Quality Check

We conducted a quality assessment through the risk of bias evaluation by adhering to The Cochrane Collaboration’s Guidelines [38]. We assessed each potential source based on the seven principles proposed by Higgins et al. [38]: (1) do not use quality scales; (2) focus on internal validity; (3) assess the risk of bias in trial results, not the quality of reporting or methodological problems that are not directly related to the risk of bias; (4) use judgment; (5) choose domains to be assessed based on theoretical and empirical considerations; (6) focus on the risk of bias in the data represented in the review rather than as initially reported; and (7) rate the outcome-specific evaluations of the risk of bias.

Furthermore, we graded each study based on the GRADE system using a four-point scale (very low [1], low [2], moderate [3], and high [4] quality). We included all studies averaging at least ‘moderate quality’ (3) based on two evaluators’ averages whose ratings should not have differed by more than one point. While two-point discrepancies did not emerge during the evaluations, we were prepared to seek the help of an independent scholar in such an eventuality, and based on her opinion, we would adjust the score. The key outcome of the quality assessment was that the reported results had to be attributable to placebo and/or nocebo effects, and the study could be easily replicated; based on reporting clarity and methodological details. Studies satisfying these two key points were rated as moderate or high quality and included in the review.

## 3. Results

### 3.1. Description of Studies

We located 103 articles on PubMed (4), Web of Science (49), Medline (4), and EBSCO (46). Additional records were also identified through other sources, like cross reference, Google Scholar, bibliography check, and manual search (18). Out of 121 articles reviewed for titles and abstracts, 25 duplicates were removed. The remaining 92 eligible studies were then assessed through full-text screening. After that, 72 articles were eliminated based on the exclusion criteria. Finally, at the end of the screening process, 20 reports were considered eligible (see Figure 2).

### 3.2. Study Design and Participants

Most studies used a within-participant design, except for three employing between-participants [21,28,34]. Fifteen studies examined the placebo and nocebo effects of nutritional ergogenic aids [18,19,20,21,23,24,26,27,28,29,31,33,34,35,36]. At the same time, four focused on mechanical ergogenic aids [22,25,30,37], and one study investigated the open-label placebo [32] (see Table 3). Finally, fifteen included studies examined healthy participants [19,20,21,22,23,24,25,26,27,29,30,31,32,33,35] (n = 397). One study sampled untrained males [28] (n = 14), another tested paralympic athletes [36] (n = 4), and two examined children; one tested elite child athletes [18] (n = 12), and another worked with children affected by obesity [34] (n = 24). Sample sizes ranged between 4 and 78. Most participants were males. Overall, there were 535 participants.

### 3.3. Nutritional Ergogenic Aids

#### 3.3.1. Caffeine

In five studies, researchers examined the placebo effects of caffeine [19,20,29,33,36]. Hurst et al. [33] found that participants ran faster in informed/received caffeine and informed caffeine/received placebo trials, which showed a comparable effect of caffeine to placebo (Cohen’s *d* = 1.14). Likewise, Costa et al. [36] revealed that caffeine placebo intake can noticeably boost both mean velocity (Cohen’s *d* = 0.36) and mean propulsive velocity (Cohen’s *d* = 0.49) in paralympic weightlifters. Finally, Valero et al. [19] noted that belief in a caffeine-containing pill boosts 6 min running performance by 1.6% (Cohen’s *d* = 0.69) without altering pacing.

However, two studies [20,29] demonstrated that placebo caffeine did not significantly alter performance observed in strength measures (Cohen’s *d* = 0.50), as indicated by Ortiz-Sanchez et al. [20]. Similarly, Filip-Stachnik et al. [29] reported that caffeine used as a placebo did not significantly enhance performance (Cohen’s *d* = 0.26); a percentage change of −1.7% demonstrated no improvement.

Therefore, it appears that the placebo effects of caffeine depend on the participants and exercise or motor performance characteristics. Studies are necessary to systematically categorize the type of performance and classify the placebo response to each by also considering the type and characteristics of the participants. Still, we conclude that in some cases, like running [33], caffeine administered as a placebo might have a comparable effect to genuine caffeine.

#### 3.3.2. Mint and Menthol Gel

In two studies, the researchers reported the placebo effect of a mint (Tic Tac) and menthol gel [18,24]. Szabo [18] showed that a Tic Tac could enhance elite child athletes’ objective kayak sprint performance through belief manipulation by the coach and a role model actor. The participants thought that one of the Tic Tac ingredients has performance-enhancing properties and, consequently, showed a 3.65% increase in distance during the 2 min kayak ergometer sprint (Cohen’s *d* = 2.8).

In contrast, Vogel et al. [24] reported that the menthol effect was negligible or trivial for the total distance covered during the 20 min time trial running performance measure in the menthol-enhanced energy gel condition (Cohen’s *d* = 0.0). No significant differences emerged, and the menthol gel was inferior to a placebo gel. Although these results contradict previous research on menthol use in sports and exercise, these findings could only be specific to the study’s sample characteristics [39] and suggest that a placebo agent could be better than an agent with active ingredients. Therefore, the form of placebo [1], participants, type of motor performance, and exercise duration might influence mint or menthol’s effects in placebo research.

#### 3.3.3. Colored Artificially Sweetened Solution

A study [23] used a pink artificially sweetened solution as a placebo to examine the ergogenic benefit on endurance exercise performance. The results showed that consuming the pink drink improved strength and endurance exercise. The effect was ascribed to the color-induced perception of sweetness and the perceived presence of carbohydrates (Cohen’s *d* = 0.40). However, the participants thought they received a carbohydrate (CHO) drink with ergogenic potential. While the drink did not contain CHO, participants still benefited from moderate ergogenic benefits.

#### 3.3.4. Energy Meals

Another study by Naharudin et al. [27] assessed the potential of carbohydrate and placebo breakfasts containing xanthan gum and low-energy flavors on the participants’ resistance exercise performance. Participants were told that both breakfasts were energy rich. The number of repetitions in back squats was significantly improved after an energy-free placebo and CHO breakfast consumption compared with the water-only controlled trial, depicting remarkable performance (Cohen’s *d* = 1.0). The participants believed that they received energy-infused breakfasts, and they performed significantly better. However, they received an energy placebo and CHO meals.

#### 3.3.5. Hyperoxic Air

Davies et al. [31] tested the impact of normoxic air (regular air concentration) and hyperoxic air (oxygen-enriched air) on 400 m cycling trials. In three trials, participants were told that they were inspiring hyperoxic air. However, they were deceived in the third trial, and they inspired normoxic air. Later, the deception was unveiled, and participants were asked to reproduce their performance. The researchers found that the participants exhibited a larger power output and quicker performances when breathing hyperoxic air in truthful and deceptive trials (8% increase). This experiment shows the conditioned placebo effect.

#### 3.3.6. Water

Fanti-Oren et al. [34] showed that a placebo effect (of positive information with ingestion of water) led to significantly higher peak heart rates and longer exercise duration in both children with obesity and normal weight children (Cohen’s *d* = 0.70, 0.90, respectively). Similar to Szabo [18], this work exemplifies how mere instruction associated with an agent could benefit performance, especially in children because they trust adults [18].

#### 3.3.7. Inert Capsules

Further, McLemore et al. [28] revealed nocebo effects in strength training for ROM (Cohen’s *d* = 1.80), RPE (Cohen’s *d* = 2.50), and total repetitions performed (Cohen’s *d* = 0.08) after having participants ingest an inert substance (gluten-free cornstarch capsule). These effects occurred in the negative belief group 48 h post-exercise.

#### 3.3.8. Sports Cream

Horvath et al. [21] reported that sports cream did not significantly affect postural stability (Cohen’s *d* = 0.10). Despite this finding, the expectations influenced subjective perception of performance.

#### 3.3.9. Equipment

Blumenstein et al. [26] showed that belief in equipment affected ski performance, with a small effect size (Cohen’s *d* = 0.15). Overall, negative perceptions slow performance.

#### 3.3.10. Verbal Suggestions in Clinical Scenarios

Zech et al. [35] presented that the weakening effect of negative suggestions results in a significant decline in muscular performance, with reductions ranging from 7–9% compared to baseline. However, when treatment benefits were highlighted, the decrease in muscle strength was mitigated, suggesting the importance of positive framing of words in communication.

### 3.4. Open Label Placebo

Saunders and colleagues [32] examined the effect of open-label placebo intervention (after ingesting a floured pill) on cycling performance and found that the results were statistically significant. There were improvements in time-to-completion in cycling performance (Cohen’s *d* = 0.16) and mean power output (Cohen’s *d* = 0.15). The results are promising. However, there was individual variation, with some improving and few remaining unchanged.

### 3.5. Mechanical Ergogenic Aids

Four included studies investigated the placebo (and nocebo) effects of mechanical ergogenic aids (transcranial direct current stimulation [tDCS]; transcutaneous electrical nerve stimulation [TENS]; Kinesio Tape [KT]) on the athletic performance of the participants [22,25,30,37].

Corsi et al. [30] examined the interplay of verbal information and conditioning in the backdrop of placebo and nocebo effects being well-informed. Participants were instructed to press a piston with maximum force in a motor task. TENS over the primary motor cortex was utilized to measure motor-evoked potentials (MEP) and the cortical silent period (CSP) from the muscle engaged in the task. The results showed that negative verbal suggestions counteracted the effects of positive conditioning, causing participants to feel weaker, exert more effort, and produce less force, mirroring strong nocebo effects (Cohen’s *d* = 1.51). The findings revealed that if the expectation was negative, the outcome was negative; however, if the expectation was positive, it reaped positive results, too. This study exemplifies the power of negative suggestions overriding conditioned effects.

Further, Fiorio and colleagues [25] found that compared to the control condition, a TENS placebo condition yielded shorter movement time, reflecting a sizeable placebo effect (Cohen’s *d* = 0.64), showing promising results.

In another study, Hanson et al. [22] demonstrated that tDCS had no placebo or nocebo effects to improve exercise aerobic performance. The participants were grouped into belief and disbelief groups and were informed about the tDCS being effective and non-effective in their performance, respectively. No significant improvement was seen compared to the disbelief group (Cohen’s *d* = 0.03).

Finally, researchers [37] highlighted the placebo effects of facilitatory KT, resulting in a small increase in grip strength among regular KT users (Cohen’s *d* = 0.16) but not for non-users. This study, again, reveals the placebo effect occurring through conditioning.

### 3.6. Type of Sport or Motor Performance

Out of 20 included studies, 4 studies measured the participants’ running performance [19,24,33,34]. One study measured the kayaking performance of elite athlete children [18]. Another six studies measured strength tasks (for instance, bench press, squat) as the performance measure [20,23,27,29,35,37]. Postural stability was also measured by researchers [21]. Roller ski sprint, 3 m aerobic test, and bench throw were assessed by three different studies [22,26,36]. Furthermore, limb movement was also measured in one study [25]. The motor task ‘press a piston’ was assessed in another study [30], while RPE and ROM were measured by two studies [28,33]. Lastly, cycling performance was examined in two studies [31,32].

### 3.7. Side Effects

Two studies reported side effects of administering caffeine as placebo [19,20]. Ortiz-Sanchez and colleagues [20] reported that participants experienced increased nervousness, extremely fast heart rate, activeness, and urine production. Due to these side effects, the performance was not enhanced. Valero et al. [19] also highlighted that ingesting a placebo (caffeine) caused side effects such as nervousness, gastrointestinal discomfort, muscular pain, or even a headache.

### 3.8. Overall Results

Twenty studies, with 535 participants, investigated placebo and nocebo effects on the participants’ sporting performance. Pooled effect sizes revealed a moderate-to-large placebo effect size across all studies (Cohen’s *d* = 0.67), suggesting that the treatments were moderately strong. It also surpassed the effect size reported in previous reviews, indicating that placebo and nocebo effects can significantly influence athletic performance, even more than previously thought.

Small to large effects were discovered for nutritional (*d* = 0.86) ergogenic aids. For mechanical ergogenic aid, the effects were moderate (*d* = 0.38), for cream and gel (*d* = 0.05), and for open-label placebo (*d* = 0.16). The nocebo effects were almost twice as large as the placebo effects as based on five measures from three studies (*d* = 1.20). The current findings support and extend the previous evaluation in the field by providing additional evidence.

## 4. Discussion

This review sought to synthesize the recent findings on placebo and nocebo effects on sports performance, building on Hurst et al. [11]. Our synthesis covers the period from January 2019 to the end of May 2024, providing an updated analysis of this evolving field. Since Hurst et al. did not include any papers from 2019, we searched papers published from January 2019. In their excellent review, Hurst et al. located 32 studies involving 1513 participants and reported moderate placebo (*d* = 0.36) and nocebo (*d* = 0.37) effects. The effects of nutritional (*d* = 0.35) and mechanical (*d* = 0.47) ergogenic aids were also small to moderate. Similarly, the pooled effect size across all studies was small and moderate (*d* = 0.38). Overall, the new studies support Hurst et al.’s earlier conclusions.

Still, the overall effect sizes are larger, but this finding is due to six studies reporting effect sizes equal to or greater than 1.0. Several studies, however, showed no or only small effects, demonstrating the high variability in the research findings. This heterogeneity is due to many factors, including the placebo or nocebo agent, cultural or social values, norms and beliefs, type of motor performance measures, and even the environment, which all yield the subjective appraisal of the certainty of the administered placebo [1] and determine their effects. Above and beyond these factors, personality factors, past learning, and life experience also determine placebo responsiveness. For a detailed discussion of these determinants and a model incorporating them, the readers are referred to Szabo [1]. Furthermore, even cultural differences might affect the placebo/nocebo response, but there is no research on this issue in the context of sports and exercise.

Placebo effects were reported in 13/20 studies [18,19,20,23,25,27,30,32,33,34,36,37], which yielded a moderate-to-large effect site (*d* = 0.67), but in one study [31], the effect size could not be determined, and two studies [30,32] offered effect sizes for more than one intervention. The mean effect size reported here is the largest ever reported for motor performance facilitated by placebo effects. However, one study [18] conducted with children, another using strength training [27], and one with runners [33] yielded large effect sizes and thus contributed to an increased mean effect size. A novelty in this review is that even an open-label placebo can induce motor performance improvement [32] despite yielding only a small effect site. Future research in this area is warranted. Another novelty is that two studies with children were located [18,34], and both reported placebo effects with large effect sizes, showing that children’s motor performance is affected by nutritional placebos like a Tic Tac mint or water supplied with deceptive information.

Nocebo effects were reported in three studies [26,28,35] and were twice as high as the placebo effects (i.e., *d* = 1.20), but one study [26] reported a small effect size (*d* = 0.15), and for another study [35] the effect size could not be determined. So, this large average effect size stems from one study [28] suggesting that if the nocebo effects are so impactful on motor performance, more research should be conducted in this area. Finally, 4/20 studies, that is 20% of the reviewed papers, reported no placebo or nocebo effects. It should be mentioned that the reproducibility of the studies reviewed here is questionable because, unless the same people are tested under the same circumstances, samples will involve people with different experiences, different ratios of placebo and nocebo responders, and perhaps performing a sport or motor performance in different conditions. Thus, while similar findings could be expected, their magnitude may vary due to the studied sample.

What none of the research presented is whether their participants were or not placebo responders. We can only speculate that the effect sizes, or the impact of placebo and nocebo interventions, would be higher in placebo/nocebo responders and lower in non-responders. We report here, along with past reviews, a sort of averaged effect size across placebo responders and non-responders. From a practical perspective, the placebo responders are more likely to benefit from placebo interventions than non-responders. Similarly, they probably experience more harm from nocebo interventions than their non-respondent counterparts [1]. Future research should aim to investigate these populations separately, especially when placebo doping is considered from a practical perspective and when negative communication with (some/responder) athletes may act as a nocebo with the potential to affect the sporting career of the individual.

### 4.1. Side Effects

While side effects of placebos are not explored explicitly in the reviewed papers, it is worth mentioning the widespread recognition of anabolic steroids as a popular ergogenic aid despite their illegality and severe health risks. Studies have linked steroid use to cardiovascular issues such as atherosclerosis and insulin resistance [40]. Similarly, amphetamines, while enhancing performance, are associated with adverse psychological effects like hallucinations, anxiety, and hypertension. Moreover, both substances carry a high risk of addiction and dependence. Using placebos instead of these substances could significantly reduce the health risks associated with their use.

Further, caffeine use in sporting performance has increased because of its ergogenic benefits [19,20]; however, it often causes side effects like insomnia and nervousness. These effects can influence an individual’s subjective experiences on objective measures.

However, unless one uses an open-label placebo, planned placebo administration involves deception, which could also result in side effects. For example, an athlete may become disappointed and lose trust in the person delivering the placebo, often the coach [9], and stream downhill on their sporting career paths. Being deceived may trigger strong emotions in those placebo responders who exhibit trust in an authoritative figure. They may feel bad about themselves, too, for trusting the individual or believing in the information associated with the placebo agent or event.

The side effects reported in two studies [19,20] in the present review should not have occurred, but the participants in these studies perceived them as real. There is a strong practical implication, especially with nocebo effects emerging as having a more potent impact than placebo effects, advancing the message that words can trigger undesired mental or psychophysiological reactions. There is a need for more research on the side effects of palcebos and nocebos in sports and exercise.

The variability of the results among the included studies suggests the need for further standardization of research methodologies in this field. Factors that need to be considered are the type and physical characteristics of the placebo agents [1] and, as discussed above, measures assessing whether the participant is a responder or non-responder need to be incorporated in placebo/nocebo research. The rationale is that a study with mostly non-responders will yield no or small effects while using the same intervention with placebo responders could yield large effects.

Most placebo research involves deception. In research settings, such studies are performed with ethical permission from a research ethics board. In applied settings, Szabo [1] asks the question, is deception always harmful? While he admits that deception is unethical, he provides examples where deception with good intent might be a means to help one achieve a personal goal. In sports, it is a form of mental doping, which, whether internal or external, concealed or open, resorts to the power of the mind to influence favorably one’s thoughts, which could facilitate performance [1], but hidden placebos should only be administered after obtaining consent from the athlete, while open placebos can be used at any time but still only with the athlete’s consent.

### 4.2. Limitations

This review is not without limitations. The included studies exhibit heterogeneity in methodologies and samples of different age groups, which complicates comparisons and the synthesis of findings. Furthermore, many studies that met our inclusion criteria had small sample sizes, limiting their statistical power and generalizability. Additionally, the context dependency of placebo and nocebo effects introduces challenges, as factors such as cultural beliefs and laboratory settings may influence the magnitude of these effects. Therefore, drawing definitive conclusions and generalizing findings to broader populations or settings is unadvised.

### 4.3. Practical Implications

This research presents implications for research and applied sports science, with treatment effect sizes ranging from moderate to large, suggesting significant potential benefits for athletes. Through this research, nutritional ergogenic aids could represent alternatives to chemical performance enhancers without posing health risks. Beyond athletes, their support networks can also leverage this knowledge by implementing (safe) placebo interventions on the field while being mindful of the expectations and beliefs of the athletes. Healthcare professionals could apply these insights to optimize treatment outcomes by cultivating positive expectations and reducing negative ones. Coaches should be attentive in their communication of constructive criticism by being aware of the possible nocebo effects of the interpretation of some words, especially by the individuals who are nocebo responders. Moreover, policymakers stand to gain from this research as it offers opportunities to develop strategies that harness placebo effects for promoting positive health behaviors or mitigating nocebo effects, thereby enhancing public health initiatives. However, to avoid deception, such work may be most fruitful with open-label placebo interventions.

## 5. Conclusions

The included studies reveal that placebo effects have small-to-large effects on various motor performances, potentially influencing sports and exercise performance. Nocebo effects might have twice as strong effects, but the effect size calculated in this review is based on three studies only. However, while the average effects emerge as relatively moderate for placebo effects and large for nocebo effects, there is a wide discrepancy between various studies ascribable to participants, type of placebos or nocebos, and outcome measures. There is a disregard for personality factors, such as being a placebo responder or not. Future work should look at these population segments with a practical consideration of who can benefit or be hurt by placebo and nocebo interventions. Overall, this five-year period review concludes that placebo and nocebo effects exist in sports and exercise, but the heterogeneity of research in the field prevents more specific conclusions.

## Figures and Tables

**Figure 1 nutrients-16-01975-f001:**
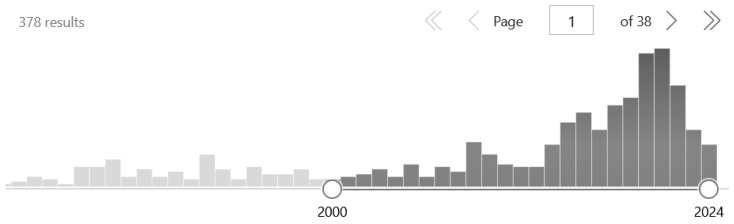
Papers having the terms ‘placebo or nocebo’ and ‘sport or exercise’ in their title in the PubMed database between 2000–2024. Bars represent the number of articles (e.g., 51 in 2020, 54 in 2021, 42 in 2022, and 23 in 2023), showing a decline from 2022.

**Figure 2 nutrients-16-01975-f002:**
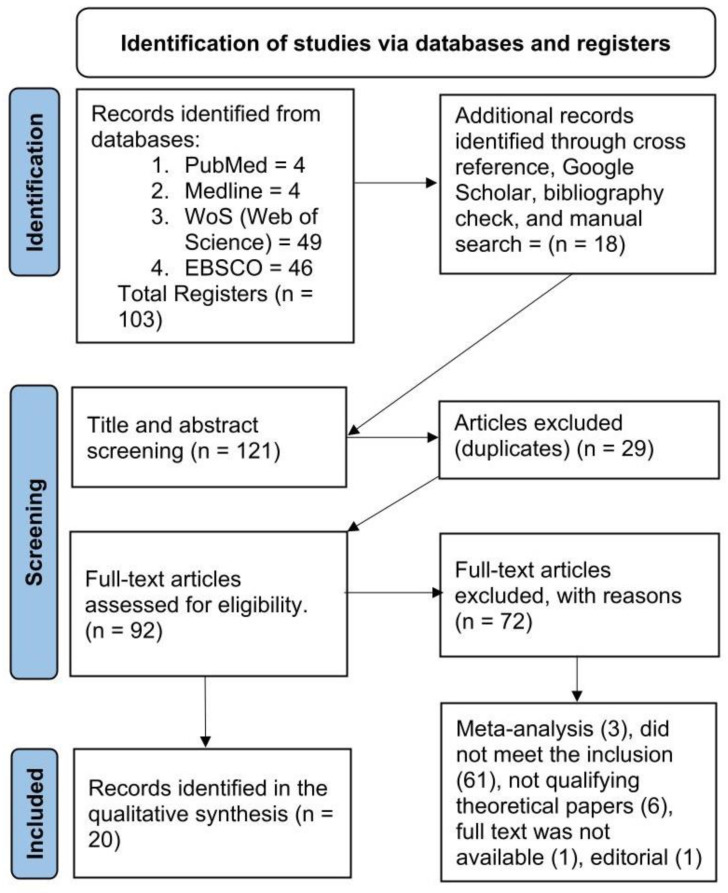
The literature searches (n = the number of records/articles).

**Table 1 nutrients-16-01975-t001:** Delimitations of the study: inclusion and exclusion criteria for the eligible articles.

Inclusion Criteria	Exclusion Criteria
Empirical research papers published in 2019–2024	Books (or chapters)
Examine placebo or nocebo effects on motor performance	Conference Proceedings
Written in English	Abstracts only
Published in peer-reviewed journals	Methodological papers
Studied ergogenic aid (e.g., nutritional or mechanical)	Editorials, commentaries, reviews
Empirical research papers reporting at least one performance measure, power output, speed, time	Dissertations, unpublished manuscripts
	Non-English publications Non-peer-reviewed articles Studies reporting subjective outcomes (case studies) Studies in which the results could be attributed to other than placebo or nocebo effects or participants could guess the intervention

**Table 2 nutrients-16-01975-t002:** Search terms used in the present review.

Placebo/Nocebo Terms		Sport/Exercise Terms
Placebo effect, OR	AND	Sport * OR
Nocebo effect, OR		Performance OR
Belief effect, OR		Motor OR
Placebo response, OR		Exercise *
Nocebo response, OR		
Deceptive, OR		
Deception, OR		
Patient expectation		

Note: The * (wild card) denotes that the search should also include the plural of the term.

**Table 3 nutrients-16-01975-t003:** Search strategy for searching the Google Scholar database.

Step 1.	Access Google Scholar website.
Step 2.	Click the left upper corner lines (3) and select “Advanced search” from the pop-down menu.
Step 3.	Enter the search terms as outlined in the following steps.
Step 4.	Enter all possible keywords (i.e., sport, sports, motor, training, etc.), which denote an outcome or dependent measure, connected with the Boolean operator ‘OR’ in the “with all of the words” field. Also, add the plural of the terms.
Step 5.	Enter all independent measure names in the “with at least one of the words” field. Place double or triple words (like “placebo effect”) within quotation marks and add the plural of the terms, too. (If the Boolean operator ‘OR’ is not used, Google Scholar will add that automatically.
Step 6.	Ensure that the groups of keywords used for dependent and independent variables of interest are separated by the Boolean operator ‘AND’.
Step 7.	Set the search period in the “Return articles dated between” field.
Step 8.	Examine records gathered in order of relevance first by limiting the search to the title of the articles, then by searching anywhere within the text.

**Table 4 nutrients-16-01975-t004:** Summary of the included studies in reverse chronological order.

Author (Year; Newest First)	Design	Subjects	Dependent Measures	No. of Part.	Treatment Informed	Treatment Received (Independent Measure)	Effect Size *d*	% Change	Major Findings
Szabo (2024) [18]	WP	Elite child kayakers (♂ = 11, ♀ = 1)	200 m kayak ergometer sprint	12	Tic Tac mint (as placebo)	Tic Tac with deception and actor participation	2.8	3.7%	Greater sprint distance in placebo condition
Valero et al. (2024) [19]	WP	Recreationally trained runners (♂ = 13)	6 min TT test	13	Caffeine (optimal dose)	Gum without caffeine	0.69	1.61%	Placebo effect boosts 6 min running performance
Ortiz-Sanchez et al. (2024) [20]	WP	Physically active participants (♂ = 12, ♀ = 6)	Bench press and squat at 3 different loads	18	Caffeine (9 mg/kg)	Placebo capsule (9 mg cellulose)	0.5 and 0.5	2.56% and 11.31%	Placebo effects observed in only 2/54 strength performance measures
Horvath et al. (2023) [21]	BP	Healthy individuals (♂ = 21, ♀ = 57)	Postural stability	78	Sports cream (as placebo; as nocebo; control)	Verbal instruction connected to the same cream	0.10	2.50%	No significant objective placebo or nocebo effects
Hanson et al. (2023) [22]	WP	Leisure exercisers (♂ = 9, ♀ = 14)	3 m aerobic test	23	tDCS	Placebo and nocebo; belief and disbelief	0.03	0.25% and −1.03%	No placebo or nocebo effect
de Salles Painelli et al. (2023) [23]	WP	Strength-training athletes (♂ = 18)	Strength endurance test bench press exercise	18	Carbohydrate energy drink (CHO)	Colored, artificially sweetened drink	0.4	6.60%	The artificially sweetened solution enhanced strength-endurance exercise performance
Vogel et al. (2023) [24]	WP	Trained runners (♂ = 8, ♀ = 6)	20 min self-paced treadmill time trial	14	Placebo or menthol-enhanced energy gel	Placebo or menthol-enhanced energy gel	0	0.00%	No effect
Fiorio et al. (2022) [25]	WP	Healthy volunteers (♂ = 10, ♀ = 14)	Reach a target fine motor movement time	24	TENS	TENS (with placebo manipulation)	0.64 0.35	CBD	In contrast to control, placebo resulted in better fine motor performance
Blumenstein et al. (2021) [26]	WP	Junior athletes (♂ = 15, ♀ = 6)	45 m indoor roller ski sprint (completion time)	21	Roller skis with low, medium, and high resistance	Equal roll resistance (medium resistance; belief deception)	0.15	−0.70%	Negative perception affects roller ski performance times
Naharudin et al. (2020) [27]	WP	Strength training athletes (♂ = 22)	Back squats and bench press	22	Energy nutrition	Water, placebo, carbohydrate	1.0	14.90%	Placebo-increased strength performance
McLemore et al. (2020) [28]	BP	Untrained males (♂ = 14)	ROM, RPE, and total repetitions performed	14	Negative belief group: inorganic nitrate	Inert substance capsule	1.8 2.5	−22.40% −49.00%	Nocebo reduced ROM and repetitions maximum performance
Filip-Stachnik et al. (2020) [29]	WP	Resistance trained (♀ = 13)	Strength endurance performance	13	Caffeine (6 mg/kg)	Flour capsule (as placebo)	0.26 (Avg.)	−1.77% (Avg.)	No placebo effect on strength performance
Corsi et al. (2019) [30]	WP	Healthy volunteers (♂ = 25, ♀ = 28)	Motor task—press a piston	53	TENS	Placebo and nocebo suggestions	0.87 and 0.67	8.12%	Suggestions affect motor performance, leading to a decline (nocebo effect)
Davies et al. (2019) [31]	WP	Trained cyclists (♂ = 15)	Cycling TT performance	15	Hyperoxic air	Normoxic air	CBD	8%	Greater mean power output and increased speedwork due to placebo
Saunders et al. (2019) [32]	WP	Trained cyclists (♀ = 28)	Cycling time trial performance and mean power output	28	Open-label placebo	Red-and-white flour capsules (100 mg)	0.16 and 0.15	0.70% and 1.5%	Open placebo modestly enhances time trial cycling performance and mean power output
Hurst et al. (2019) [33]	WP	Trained middle-distance athletes (♂ = 11)	1000 m running TT, peak HR, RPE	11	Caffeine (3 mg/kg)	Told caffeine, received placebo	1.14	1.89%	Large placebo effect compared to baseline
Fanti-Oren et al. (2019) [34]	BP	Children with obesity (♂ = 24) and normal weight children (♂ = 24)	Progressive treadmill exercise test	48	Water with the information ‘increases energy’	Water	0.71 0.90	33.3% 24.6%	Placebo-related increased time to exhaustion in both groups
Zech et al. (2019) [35]	WP	Healthy people	Maximal muscular strength	46	Negative and positive verbal and non-verbal suggestions	Positive and negative information	CBD	Around 7% to 9% decrease	Nocebo effects weakened muscular performance
Costa et al. (2019) [36]	WP	Paralympic weightlifting athletes (♂ = 4)	Bench throw tests (mean propulsive velocity)	4	Caffeine capsule (6 mg/kg)	Maize starch capsule (as placebo)	0.36 0.49	10.5% 13.5%	Placebo improved bench-throw performance
Mak et al. (2019) [37]	WP	KT users and non-KT users (♂ = 32, ♀ = 28)	Maximal power grip strength	60	KT (as placebo)	Facilitatory KT or no tape application (control)	0.16	0.16%	KT increases grip strength slightly

Note: Only the placebo group or condition versus a control or baseline measure is considered in the Tables. Avg. = average; BP = between participants; CBD = cannot be determined; *d* = effect size (Cohen’s *d*); No. of Part. = number of participants; ROM = range of motion; tDCS = transcranial direct current stimulation; WP = within participants; ♂ = males and ♀ = females; TENS = transcutaneous electrical nerve stimulation; TT = time trial; HR = heart rate; RPE = rating of perceived exertion; KT = Kinesio Tape.

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
