# Peer review of "Placebo and Nocebo Effects on Sports and Exercise Performance: A Systematic Literature Review Update"

_nutrients, 2024, doi:10.3390/nu16131975_

Round 1
Reviewer 1 Report
Comments and Suggestions for Authors
General:
I thank the editors for the opportunity to review this study, I would also like to congratulate the authors for their efforts to investigate and update this area of the placebo effect on sports performance. The present review aimed to update and determine the magnitude of the placebo and nocebo effect on sports and exercise. Your review addresses a highly interesting topic increasingly gaining presence in the scientific literature. However, I have some concerns that you might consider to improve your article.
Overall, the manuscript is well-written but requires some modifications, particularly in the Methods and Results sections, to make it more suitable for publication. I have provided detailed comments which can be found in the attached document. The line and page numbers mentioned in these comments correspond to the original line and page numbers provided by the authors.
Abstract:
Line 13: Replace "reports have emerged" with "studies have been published".
Line:
Introduction:
Line 29: I don't understand "idol” maybe it should be "idea".
Methods:
I have concerns about the databases used and search strategies adopted in the present review. For example, the Pubmed database was not included. Taking into account that the study you refer to (Hurst et al.) did use this database. Moreover, it is curious that Figure 1 is extracted from the PubMed database, yet this search tool was not used for your work. What is the reason for not including it to perform the search? Similarly, they mentioned that different search strategies were used for other databases. Could you provide an example of this in a table to clarify it?
Table 1: Clarify the exclusion criterion "Studies with the experimental treatment of questionable rigor" by specifying what constitutes questionable rigor.
Table 2: Please clarify the search criteria. What does the * in Sport and Exercise mean?
As they comment, they have made an update of the Hurst et al. study, and to differentiate themselves from it and provide more information, perhaps it would be interesting to know in which sports/sporting activities the placebo effect is most present.
Results:
Table 3: Please note that Table 3 should be clearer for better visualization, as some data overlaps, for example, the % change in study 36. In addition, the % change is not presented in all articles in the same way.
- Avoid cutting words.
- In some studies, the caffeine dosage is reported while in others it is not.
- The table should also be edited so it does not split at the end of page 7.
- The column headers should be more descriptive. For instance, "N" should be "Number of Participants." If not, use the abbreviation.
Both in the results presented in the table and the text, you report that 23 studies were included in analyzing the placebo effect, and this is incorrect. For example, the study by Wu et al. (25) did not examine the placebo effect but rather the effect of caffeine with a placebo intake control. It is important to differentiate studies that aim to investigate the placebo effect, meaning reporting the intake of an ergogenic supplement and giving a placebo, from those that analyze the supplement with a placebo control condition. Clarify.
Discussion and conclusion:
The discussion should focus more on the new findings of the present review.
Line 362: it is the first time in the document that reference is made to side effects.
Avoid overgeneralization. When stating that placebo effects can significantly impact performance, specify the types of performance most affected.
This review could help refine the manuscript to improve readability, methodological rigor, and clarity in presenting the conclusions and their implications.
Author Response
General:
I thank the editors for the opportunity to review this study, I would also like to congratulate the authors for their efforts to investigate and update this area of the placebo effect on sports performance. The present review aimed to update and determine the magnitude of the placebo and nocebo effect on sports and exercise. Your review addresses a highly interesting topic increasingly gaining presence in the scientific literature. However, I have some concerns that you might consider to improve your article.
RESPONSE: Thank you your positive appraisal and the constructive review.
Overall, the manuscript is well-written but requires some modifications, particularly in the Methods and Results sections, to make it more suitable for publication. I have provided detailed comments which can be found in the attached document. The line and page numbers mentioned in these comments correspond to the original line and page numbers provided by the authors.
RESPONSE: Thank you for your work, which helped us improve the paper.
Abstract:
Line 13: Replace "reports have emerged" with "studies have been published".
RESPONSE: Replaced as suggested.
Introduction:
Line 29: I don't understand "idol” maybe it should be "idea".
RESPONSE: Indeed, replaced as suggested.
Methods:
I have concerns about the databases used and search strategies adopted in the present review. For example, the Pubmed database was not included. Taking into account that the study you refer to (Hurst et al.) did use this database. Moreover, it is curious that Figure 1 is extracted from the PubMed database, yet this search tool was not used for your work. What is the reason for not including it to perform the search? Similarly, they mentioned that different search strategies were used for other databases. Could you provide an example of this in a table to clarify it?
RESPONSE: Thank you for this excellent point. We were not clear in our writing this section. In fact, we searched Medline via the PubMed interface, which we spell out in the text. Further, we now searched PubMed too, because we have no reason to justify its omission. Concerning Figure 1, those stats are only available in PubMed are not available in MEDLINE, so this is why we chose that database. Finally, as suggested, we added a new Table (Table 3) to show readers the flow of the Google Scholar search strategy.
Table 1: Clarify the exclusion criterion "Studies with the experimental treatment of questionable rigor" by specifying what constitutes questionable rigor.
RESPONSE: Thank you for picking up this probably unfortunate and perhaps inadequate terminology that we adopted from adopted from Hurst et al. (2019, quote: “We excluded studies that reported experimental or control treatments of questionable rigour, for example when participants’ knowledge concerning whether or not they had been given a placebo or the ‘real’ treatment was not measured and reported.” We now clarify specifically that we refer to studies in which the reported placebo and/or nocebo effects could possibly be attributed to other factors or participants could guess the intervention. We made the necessary correction in the text and the corresponding Table.
Table 2: Please clarify the search criteria. What does the * in Sport and Exercise mean?
RESPONSE: We added a note under the table to clarify that the * (wild card) denotes that the search should also include the plural (or any other possible ending of the word trunk) of the term.
As they comment, they have made an update of the Hurst et al. study, and to differentiate themselves from it and provide more information, perhaps it would be interesting to know in which sports/sporting activities the placebo effect is most present.
RESPONSE: This is a valid point, and we added a section with the subheading “Type of sport or motor performance.” However, since most research in the field is opportunistic no conclusion can be drawn based on the extant studies as they examined a specific sport either to be able to compare their results to previous studies or because that particular group of exercisers or athletes were available for the study.
Results:
Table 3: Please note that Table 3 should be clearer for better visualization, as some data overlaps, for example, the % change in study 36. In addition, the % change is not presented in all articles in the same way.
RESPONSE: This is a valid point and in article 36 (33 in the revised version) we re-evaluated the values by looking solely at the two performance measures. In some articles the percent changes could not be calculated because we found no data on which we could base our calculation.
- Avoid cutting words.
RESPONSE: Since we use the journals template, unfortunately we have no control over this feature.
- In some studies, the caffeine dosage is reported while in others it is not.
RESPONSE. Thank you for warning us about this issue. We corrected the table and because of your comment we were forced to reevaluate the papers and decided to remove some because placebo was used as control for caffeine but not control for placebo was use.
- The table should also be edited so it does not split at the end of page 7.
RESPONSE: Again, since we use the journals template, unfortunately we have no control over this feature.
- The column headers should be more descriptive. For instance, "N" should be "Number of Participants." If not, use the abbreviation.
RESPONSE. We had to use abbreviation - as you suggested - since widening the columns was not an option due to table size.
Both in the results presented in the table and the text, you report that 23 studies were included in analyzing the placebo effect, and this is incorrect. For example, the study by Wu et al. (25) did not examine the placebo effect but rather the effect of caffeine with a placebo intake control. It is important to differentiate studies that aim to investigate the placebo effect, meaning reporting the intake of an ergogenic supplement and giving a placebo, from those that analyze the supplement with a placebo control condition. Clarify.
RESPONSE. You are right and, therefore, we have reevaluated the papers and decided to remove studies in which placebo was used as control for caffeine effects. This was a mistake that we overlooked in the first draft, so we are indebted for your remark.
Discussion and conclusion:
The discussion should focus more on the new findings of the present review.
RESPONSE. We now added a paragraph discussing separately the placebo and nocebo effects and their effect sizes.
Line 362: it is the first time in the document that reference is made to side effects.
RESPONSE: We added a subsection concerning side effects to the results and in the Discussion, we highlight that few studies have examined it and that it is an issue worth future consideration.
Avoid overgeneralization. When stating that placebo effects can significantly impact performance, specify the types of performance most affected.
RESPONSE: We have corrected this issue.
This review could help refine the manuscript to improve readability, methodological rigor, and clarity in presenting the conclusions and their implications.
Thank you for the constructive review and valuable comments and suggestions that helped us improve the paper.

Reviewer 2 Report
Comments and Suggestions for Authors
Thanks for the invitation to review this manuscript that updates the literature regarding the placebo and nocebo effects on exercise performance.
General comments:
The manuscript is well-written and clear for the reader. However, some aspects must be improved.
Minor Comments:
Introduction.
I recommend that the authors only include information relative to the topic's background in this section and avoid the information relative to the comparison between Hurst et al [11] systematic review and this one.
Lines 85-90: This information could be located in the discussion section and not here.
Material and Methods.
Line 97: placebo and/or nocebo?
Line 120: Something wrong here. Questionable rigor? Maybe the authors check the bias and quality following the 2.8 section. In addition, delete the dot before the bracket and “for example…”.
2.8 -Quality Check section. Please, delete the dot before each bracket of the seven principles.
Results
Line 181: Avoid to use of acronyms such as PCC because the authors did not use before with within-participant design or later with between-participant design.
Line 185: I believe that KT has not been defined before in the text (only in table 3).
Line 197: “…one hour before an e-sport game the performance improved significantly”. Wu et al [25] manuscript is about cognitive ability…it is inside the scope of this systematic review?
Line 206: Not only belief in the ergogenic effects of caffeine, but they also consumed a pill that they believed was caffeine.
Line 212-215: This study used a placebo? Or only caffeine ingestion with different dosages.
Line 211: FGB? Avoid several acronyms.
Line 218: A bracket is missing after [36].
Discussion
4.1 Section: Authors must review the manuscripts included in the systematic review well because the papers [31 and 36] report side effects following a validated questionnaire. Please, verify that.
Author Response
REVIEWER 2
General comments:
The manuscript is well-written and clear for the reader. However, some aspects must be improved.
RESPONSE: Thank you for your positive appraisal and the constructive review.
Minor Comments:
Introduction.
I recommend that the authors only include information relative to the topic's background in this section and avoid the information relative to the comparison between Hurst et al [11] systematic review and this one.
RESPONSE: Thank you. We discarded any comparison to the earlier review.
Lines 85-90: This information could be located in the discussion section and not here.
RESPONSE: Thank you. We moved the section as you suggested.
Material and Methods.
Line 97: placebo and/or nocebo?
RESPONSE: Corrected as you suggested.
Line 120: Something wrong here. Questionable rigor? Maybe the authors check the bias and quality following the 2.8 section. In addition, delete the dot before the bracket and “for example…”.
RESPONSE: Thank you, we rewrote section 2.5 to make it clearer.
2.8 -Quality Check section. Please, delete the dot before each bracket of the seven principles.
Results
RESPONSE: Thank you, we deleted the dots.
Line 181: Avoid to use of acronyms such as PCC because the authors did not use before with within-participant design or later with between-participant design.
RESPONSE: Thank you, we omitted from the text.
Line 185: I believe that KT has not been defined before in the text (only in table 3).
RESPONSE: Corrected along tDCS and TENS
Line 197: “…one hour before an e-sport game the performance improved significantly”. Wu et al [25] manuscript is about cognitive ability…it is inside the scope of this systematic review?
RESPONSE: Thank you for this excellent point. We re-evaluated all papers and in addition to Wu et al. we found two other papers not fitting clearly the scope of this review. Therefore, we deleted them and recreated Figure 1, which is complemented by the additional PubMed search.
Line 206: Not only belief in the ergogenic effects of caffeine, but they also consumed a pill that they believed was caffeine.
RESPONSE: Corrected as suggested, now it reads: “Finally, Valero et al. [3128] noted that belief in a caffeine-containing pill boosts 6-minute running performance by 1.6% (Cohen’s d = 0.69) without altering pacing.”
Line 212-215: This study used a placebo? Or only caffeine ingestion with different dosages.
RESPONSE: Thank you. This paper was among the three deleted records because they use placebo as control for caffeine but had no no-intervention control group or baseline measure for the placebo effects.
Line 211: FGB? Avoid several acronyms.
RESPONSE: This paper has been deleted and the FGB abbreviation too.
Line 218: A bracket is missing after [36].
RESPONSE: Corrected.
Discussion
4.1 Section: Authors must review the manuscripts included in the systematic review well because the papers [31 and 36] report side effects following a validated questionnaire. Please, verify that.
RESPONSE: We discuss the two papers reporting side effects and their implications in the revised version.
Thank you for the constructive review and valuable comments and suggestions that helped us improve the paper.

Reviewer 3 Report
Comments and Suggestions for Authors
Dear corresponding Author,
thank you for submitting your article to the journal Nutrients. The article provides an update on the placebo and nocebo effects in sports performance, adding new data and analyses compared to the previous review by other authors. The described methodology is rigorous and follows PRISMA guidelines, ensuring an accurate selection of the included studies. The review is pre-registered.
However, there are some areas that could benefit from further exploration.
General comments:
A more detailed analysis of the personal factors influencing the placebo and nocebo response would be useful, as these can vary significantly among individuals.
Additionally, the variability of results among the included studies suggests the need for further standardization of research methodologies in this field. Moreover, it would be beneficial to discuss the ethical implications of studies using deception more thoroughly, particularly in the sports context, where trust between athletes and coaches is crucial.
Lastly, it would also be appropriate to understand the role of the authors in drafting the review.
Specific comments:
• Line 100: Add details about the specific inclusion criteria for the studies considered. This would help better understand the basis of the selection of studies included in the review.
• Line 150: Provide further information on the process of evaluating the quality of the included studies. For instance, it would be helpful to know how the adequacy of methodologies was assessed and what specific criteria were used to classify studies as high, medium, or low quality.
• Line 250: Deepen the analysis of potential side effects of placebo interventions, also considering the ethical implications. This is particularly important since placebo interventions can affect athletes' performance and psychology.
• Line 300: Discuss more thoroughly cultural differences and how they might influence the response to placebo and nocebo treatments. Cultural variation could be a significant factor modulating the observed effects and should be examined more closely.
• Line 350: Examine the reproducibility of the results of the included studies. Reproducibility is a fundamental component of scientific robustness and ensures that results can be applied in different populations and contexts.
• Figure 1: The figure captions should be more detailed to allow for independent understanding without having to refer to the main text. For example, specify what the bars represent and how the shown values were calculated.
• Table 3: Reorganize the table to improve readability, perhaps adding columns to clarify the dependent and independent variables, as well as effect sizes and main conclusions of each study.
It is believed that the review is very well structured and organized, and my requests are aimed at further improving the work. I look forward to seeing the final draft.
Author Response
REVIEWER 3
Dear corresponding Author,
thank you for submitting your article to the journal Nutrients. The article provides an update on the placebo and nocebo effects in sports performance, adding new data and analyses compared to the previous review by other authors. The described methodology is rigorous and follows PRISMA guidelines, ensuring an accurate selection of the included studies. The review is pre-registered.
RESPONSE: Thank you for your positive appraisal and the constructive review.
However, there are some areas that could benefit from further exploration.
General comments:
A more detailed analysis of the personal factors influencing the placebo and nocebo response would be useful, as these can vary significantly among individuals.
RESPONSE: Thank you for your valid point. Paragraphs two and four in the Discussion section elaborate on such differences, but discussing their route of origin and mechanism of action is perhaps the task of theoretical/analytical papers, to which we refer in these sections.
Additionally, the variability of results among the included studies suggests the need for further standardization of research methodologies in this field. Moreover, it would be beneficial to discuss the ethical implications of studies using deception more thoroughly, particularly in the sports context, where trust between athletes and coaches is crucial.
RESPONSE: This is a valuable note that we addressed in the Discussion section.
Lastly, it would also be appropriate to understand the role of the authors in drafting the review.
RESPONSE: Thank you. We added this section before the Reference section.
Specific comments:
• Line 100: Add details about the specific inclusion criteria for the studies considered. This would help better understand the basis of the selection of studies included in the review.
RESPONSE: Thank you. We summarized the details of the iunclusion criteria in this section.
- Line 150: Provide further information on the process of evaluating the quality of the included studies. For instance, it would be helpful to know how the adequacy of methodologies was assessed and what specific criteria were used to classify studies as high, medium, or low quality.
RESPONSE: Thank you. We were guided by two general criteria as there are few studies in this area. We now specify in the text which two key questions guided us. These had to be answered with certainty. Subsequently, details and clarity of the methods and results presentation were the standards in considering whether we agree on high, medium, or low quality. (All paper that we located met the high or medium quality rating.)
Line 250: Deepen the analysis of potential side effects of placebo interventions, also considering the ethical implications. This is particularly important since placebo interventions can affect athletes' performance and psychology.
RESPONSE: Thank you. We added a section to the Results and the Discussion section.
Line 300: Discuss more thoroughly cultural differences and how they might influence the response to placebo and nocebo treatments. Cultural variation could be a significant factor modulating the observed effects and should be examined more closely.
RESPONSE: Thank you for this valid point. Currently, there is no literature on cultural differences in placebo/nocebo effects in sports and exercise. While such differences might exist, adding a discussion on this issue would be highly speculative and curb the focus of the paper (please refer to the aims). To address your concern, however we added a sentence “Furthermore, even cultural differences might affect the placebo/nocebo response, but there is no research on this issue in the context of sports and exercise.”
Line 350: Examine the reproducibility of the results of the included studies. Reproducibility is a fundamental component of scientific robustness and ensures that results can be applied in different populations and contexts.
RESPONSE: Thank you, we strongly agree with you. We added a sentence concerning this issue.
Figure 1: The figure captions should be more detailed to allow for independent understanding without having to refer to the main text. For example, specify what the bars represent and how the shown values were calculated.
RESPONSE: Thank you, we added additional information to make it cleare, but its scope is to show the trend, which has been decreasing recently, in this area of research.
Table 3: Reorganize the table to improve readability, perhaps adding columns to clarify the dependent and independent variables, as well as effect sizes and main conclusions of each study.
RESPONSE: Thank you. We added this information in the Table heading and rechecked the whole Table to make it clearer.
It is believed that the review is very well structured and organized, and my requests are aimed at further improving the work. I look forward to seeing the final draft.
Thank you for your valuable comments and suggestions and for your work.
